# The advantages of abdominal compression with shallow breathing during left-sided postmastectomy radiotherapy by Helical TomoTherapy

Chalardchay Pratoomchart[1], Pitchayaponne Klunklin[1,2]*, Somsak Wanwilairat[1], Wannapha Nobnop[1,2], Kittikun Kittidachanan[1], Imjai Chitapanarux[1,2]

1 Division of Radiation Oncology, Faculty of Medicine, Chiang Mai University, Chiangmai, Thailand,
2 Northern Thai Research Group of Radiation Oncology (NTRG-RO), Faculty of Medicine, Chiang Mai University, Chiang Mai, Thailand

* pitchayaponne.kl@cmu.ac.th

## Abstract

### Background

Left-sided post-mastectomy radiotherapy (PMRT) certainly precedes some radiation dose to the cardiopulmonary organs causing many side effects. To reduce the cardiopulmonary dose, we created a new option of the breathing adapted technique by using abdominal compression applied with a patient in deep inspiration phase utilizing shallow breathing. This study aimed to compare the use of abdominal compression with shallow breathing (ACSB) with the free breathing (FB) technique in the left-sided PMRT.

### Materials and methods

Twenty left-sided breast cancer patients scheduled for PMRT were enrolled. CT simulation was performed with ACSB and FB technique in each patient. All treatment plans were created on a TomoTherapy planning station. The target volume and dose, cardiopulmonary organ volume and dose were analyzed. A linear correlation between cardiopulmonary organ volumes and doses were also tested.

### Results

Regarding the target volumes and dose coverage, there were no significant differences between ACSB and FB technique. For organs at risk, using ACSB resulted in a significant decrease in mean (9.17 vs 9.81 Gy, p<0.0001) and maximum heart dose (43.79 vs 45.45 Gy, p = 0.0144) along with significant reductions in most of the evaluated volumetric parameters. LAD doses were also significantly reduced by ACSB with mean dose 19.24 vs 21.85 Gy (p = 0.0036) and the dose to 2% of the volume (D2%) 34.46 vs 37.33 Gy (p = 0.0174) for ACSB and FB technique, respectively. On the contrary, the lung dose metrics did not show any differences except the mean V5 of ipsilateral lung. The positive correlations were found

**Data Availability Statement:** All relevant data are within the paper and its Supporting Information files.

**Funding:** The authors received no specific funding for this work.

**Competing interests:** The authors have declared that no competing interests exist.

between increasing the whole lung volume and mean heart dose (p = 0.05) as well as mean LAD dose (p = 0.041) reduction.

## Conclusions

The ACSB technique significantly reduced the cardiac dose compared with the FB technique in left-sided PMRT treated by Helical TomoTherapy. Our technique is uncomplicated, well-tolerated, and can be applied in limited resource center.

## Introduction

Many post-mastectomy radiotherapy (PMRT) trials [1] showed an increase in both disease-free survival and overall survival by almost 10%, regardless of the size of the primary tumor or the number of lymph node involvement. Consequently, clinical practice guidelines] [2–4] endorses PMRT for patients who had four or more involved axillary lymph nodes and strongly recommended for one to three positive nodes [2].

During radiation in breast cancer, the adjacent normal structures receive radiation doses which contributed to acute and late complications. Especially in left-sided breast cancer (LtBC) which the treatment inevitably leads to a radiation dose to the critical organs, i.e., heart, ipsilateral lung, and contralateral lung, causing cardiac and pulmonary function disorders in the long-term [5–7].

To balance the risk and benefit of radiation in LtBC, there are many attempts to limit the radiation dose to cardiopulmonary organs. Modern radiotherapy techniques such as three-dimentional, forward-planning IMRT, inverse-planning IMRT and rotational IMRT were developed to decrease the radiation dose to the heart and lungs in breast cancer radiotherapy [8]. However, the more famous option for cardiac and lung-sparing is using the application to assess respiratory organ motion, such as a breathing-adapted technique currently used to reduce inter and intrafractionation motion and also allow for organs at risk (OARs) dose reduction [9].

The breathing adapted technique applied in LtBC radiotherapy was divided into two different processes, one is a selection to treat in specific respiratory phase by using a gating system while patients do a free breathing and the other is a deep-inspiration breath-hold (DIBH) technique which allowed patients voluntary deep-inspiration breaths or used active-breathing control (ABC). Among these two processes, the DIBH gained more popularity with many published data with a significant reduction in the amount of radiation hitting the heart, lungs and other OARs and decreasing the future risk of heart disease [10–19].

Abdominal compression was initially established for stereotactic body radiation therapy (SABR) of lung and upper gastrointestinal tumors [20, 21]. This technique uses a plate pressed against the patient's abdomen for minimizing respiratory motion. Some report found the benefit of using abdominal compression in hepatocellular carcinoma and cholangiocarcinoma who received liver SABR by lowering the dose to the liver [20].

Tomotherapy treatment planning can be classified into two modes, TomoHelical (HT) and TomoDirect (TD). The HT plans showed better target coverage and OARs sparing for the chest wall and regional nodal radiation with higher plan quality scores when compared with TD plans [22]. In our center, large numbers of post-mastectomy breast cancer patients required radiation to the chest wall and whole regional lymph node area. Therefore, for this group of patients, HT is always prescribed. Regarding HT mode planning, the flash function

which is the extending of the treatment field to compensate for the respiratory motion was limited. To reduce the chest wall movement due to respiration for ensuring the perfect target dose coverage during treatment and effort to conduct lung and cardiac sparing along with a cover the respiratory movement problem, we hypothesized the usage of our abdominal compressor in LtBC PMRT.

The DIBH technique provides benefit in reducing the cardiopulmonary dose by doing deep inspiration and hold. This breathing maneuver will increase the lung volume and the distance between chest wall and heart by flattening of diaphragm and expansion of the thorax. Without beam gating system to complete DIBH procedure in our center, we try to apply abdominal compression with a patient in the deep inspiration phase and carry on the shallow breathing to get the same benefits. Therefore, this study aimed to evaluate the advantage of using abdominal compression with shallow breathing (ACSB) compared to the free breathing (FB) technique in decreasing cardiopulmonary radiation dose (heart, Left anterior descending artery (LAD), Ipsilateral lung, contralateral lung in the PMRT of LtBC. Hopefully, our results will provide an alternative option to minimize cardiopulmonary radiation dose in LtBC PMRT.

## Materials and methods

### Patients

The Faculty of Medicine, Chiangmai university Research Ethics Committee reviewed and approved this study (RAD-2562-06551). The writing consent was obtained from all patients willing to participate in our protocol. Between December 2019 and August 2020, potentially eligible patients seen in consultation in our radiation oncology clinic were approached by a research team for consideration of the study. Inclusion criteria were the following: ages between 18–70 years, newly diagnosed with invasive ductal or lobular carcinoma of left breast, non-metastatic disease, underwent left mastectomy with axillary lymph node dissection and planned to receive adjuvant irradiation to chest wall and regional nodal areas. We excluded patients with carcinoma in situ and pathology other than invasive ductal or lobular carcinoma or who have prior radiotherapy to chest area. The patients who were unable to inspire adequately or deeply or unable to tolerate abdominal compression were also excluded from our protocol.

### Immobilization and simulation

Non-contrast CT simulation (Siemens SOMATOM Definitions AS 64 slices) was performed with the patient in a supine position with both arms extended above the head using the wing board. Radiopaque wires were placed on the patients' skin to define scars and field borders. The scan was acquired from the level above the cricoid cartilage through five centimeters below the clinically marked inferior port edge of the chest wall with 3-mm slice thickness. The entire lungs must be included. Each patient was simulated by two techniques, free-breath (FB) and abdominal compression with shallow breathing (ACSB) technique. Two CT simulations were set separately to decrease the carry over effect (one in the morning and the other in the afternoon). For the ACSB technique, we trained patients to do the deep inspiration then hold their breath to prepare for proper breathing during the procedure. Before performing the CT scan, the patients needed deep inspirations with the Anzai belt respiratory gating system (AZ-733VI Rev.1.0 by Ansai Medical, Co., Ltd., Shinagawa-Ku, Japan) to monitor the respiratory signals. The abdominal compression (ONEBridge™ by CIVCO Radiotherapy, Orange City, IA, USA) was applied as the patient tolerated. After that, we let the patient breath normally under abdominal compression to create shallow breathing. When the respiratory cycle graph showed

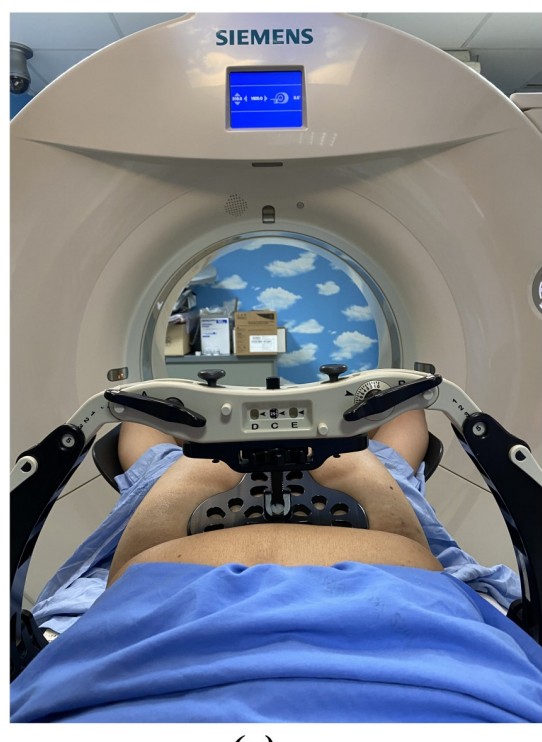

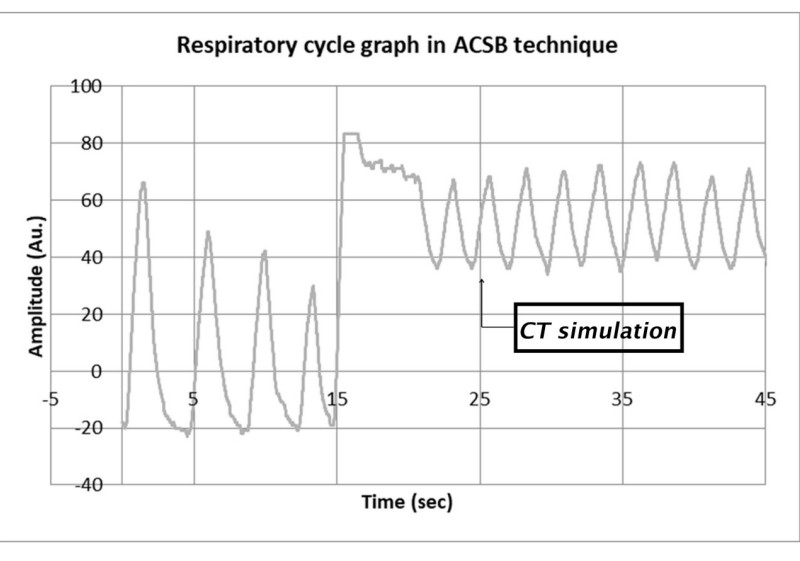

**Fig 1.** Compression devices in ACSB Technique (a) and Respiratory Cycle Graph (b).

a normal respiratory curve (Fig 1) which indicated that the patient was breathing regularly, the CT scan was undertaken.

## Target delineation and dose prescription

After the CT simulation, all CT images were registered in the Oncentra Master Plan (Elekta, Sweden) contouring system. The contouring was done by a single radiation oncologist. The clinical target volume (CTV) was defined as chest wall and mastectomy scar with locoregional lymph nodes. The planning treatment volume (PTV) was created by adding a 5-mm margin from CTV in all directions. The OARs consisting of the heart, left anterior descending artery (LAD), ipsilateral lung, contralateral lung and spinal cord were contoured according to the Breast Cancer Atlas for Radiation Therapy Planning: Consensus Definitions from Radiation Therapy Oncology Group [23]. The prescription dose to cover the PTV was 50 Gy in 25 fractions.

## Treatment planning evaluation

All treatment plans were created by one physicist on a TomoTherapy Hi-Art planning station version 5.1.1.6. A field width of 2.5 cm, a pitch of 0.215, and a modulation factor of 3.2 were used without bolus applied on skin. Each patient's plan was produced with the FB and ACSB techniques and evaluated according to our center's protocol (S1 Table) which were derived from ICRU report, RTOG and EORTC consensus guideline [23–25].

A plan was considered acceptable if at least 50% of the PTV received the prescription dose (D50 ≥ 50 Gy) and at least 95% of the prescribed dose (47.5 Gy) covered at least 95% of the PTV (V47.5 ≥ 95%). The hot spot defined as 115% of the prescribed dose (57.5 Gy) covered

less than 2% of the PTV volume (V57.5 ≤ 2%). For each OAR, at least one of the mandatory constraints needed to meet the criteria. The target volume constraints for the PTV and OAR dose objectives are summarized in the S1 Table. Treatment plans from the ACSB technique were explored in this study, all of our patients were treated with the FB technique.

## Statistical analysis

The Sample size was calculated based on a paired t-test analysis using following formula [26].

$$n = \frac{(Z_{\alpha/2} + Z_\beta)^2 2\sigma^2(1 - \rho)}{\delta^2}$$

The symbols are described as $Y_{control}$: outcome ($D_{mean}$) of control

$Y_{Intervention}$: outcome ($D_{mean}$) of intervention

$\mu_{control}$: mean of $Y_{control}$

$\mu_{Intervention}$: mean of $Y_{Intervention}$

$\sigma$: Standard deviation of outcome

$\rho$: Correlation between $Y_{control}$ and $Y_{Intervention}$

$\delta$: $\mu_{control}$-$\mu_{Intervention}$

Although having prior information of mean dose, $\delta$, and $\sigma$ from previous study of Schönecker et al. [10], we have no information of $\rho$. As we believed that the correlation should be more than 0.5 because we planned to measure the radiation doses in the same individual. We calculated the sample size with $\rho$ from 0.5–0.8. Finally, we considered that the sample size of this study would be maximal between the two OARs (heart and lung) which was 21 cases (using $\rho$ = 0.5).

We collected the following data, PTV volumes, cardiopulmonary organ volumes, PTV dose, and cardiopulmonary radiation doses. All data were analyzed and reported as mean ± standard deviation (SD). A Paired t-test and Wilcoxon signed ranks test were applied to estimate the statistical significance of the differences between groups, as appropriate. The Pearson correlation coefficient was used to find a linear correlation between cardiopulmonary organ volumes and doses. The results were considered to be statistically significant for P-value < 0.05. Statistical analysis was performed using SPSS statistical software (version 11.5, SPSS Inc., 444 N. Michigan, Chicago, IL, USA) and Microsoft Office Excel 2010 (Microsoft Corp. Redmond, WA).

## Results

Twenty patients with the median age of 55 (range 40–68 years) were prospectively included in the study. All patients were women and the majority of them were diagnosed with stage IIB (40%). For the ACSB technique, most patients tolerated 10 to 12 centimeters compression. All of the patients underwent left modified radical mastectomy without reconstruction and had positive lymph nodes from pathology reports, so they received radiotherapy to the chest wall, axillary lymph nodes level 1–3, supraclavicular, and internal mammary lymph node area. The characteristics of the patients were described in Table 1.

### Target volumes and doses

The PTV volumes were identical with mean PTV volume 817.55 ml (724.77–910.33 ml) for ACSB and 809.29 ml (714.73–903.85 ml) for FB technique (p = 0.2124). The use of ACSB did not compromise target coverage as indicated by a similar dose of D50% (p = 0.8761), V47.5 (p = 0.0929) and V57.5 (p = 0.5236).

## Heart and LAD volumes and doses

A physiologic decrease in heart volume was noted with the ACSB technique (576.48 vs. 603.58 ml, p = 0.0072). Moreover, using ACSB resulted in a significant decrease in the mean of the mean heart dose (MHD) with 9.17 vs 9.81 Gy (p<0.0001) and the mean of the maximum heart dose with 43.79 vs 45.45 Gy (p = 0.0144). Other evaluated volumetric parameters (V30, V25, V20, V15 and V10) were also significantly decreased as compared to the FB technique excluding the V5 as shown in Table 2.

Even though there was no statistical difference in the LAD volume, we still observed statistically significant decrease in the mean LAD dose and the dose to 2% of the volume of LAD (D2%) with the ACSB technique (19.24 vs 21.85 Gy, p = 0.0036 and 34.46 vs 37.33 Gy, p = 0.0174).

**Table 1. Baseline characteristics.**

| Variable | Value (N = 20) |
|---|---|
| Median Age (range) | 55 (40–68) |
| **Body mass index (BMI)** | |
| 18.5–22.9 | 30% |
| 23.0–24.9 | 15% |
| 25.0–29.9 | 50% |
| > 30 | 5% |
| **Staging** | |
| 2B | 40% |
| 3A | 35% |
| 3B | 5% |
| 3C | 20% |
| **T stage** | |
| T1 | 10% |
| T2 | 55% |
| T3 | 30% |
| T4 | 5% |
| **N stage** | |
| N0 | 0% |
| N1 | 60% |
| N2 | 20% |
| N3 | 20% |
| **Histology** | |
| Ductal carcinoma in situ | 0% |
| Invasive ductal carcinoma | 100% |
| **Histologic grade** | |
| Grade 1 | 0% |
| Grade 2 | 50% |
| Grade 3 | 50% |
| **Systemic treatment** | |
| **Chemotherapy** | 100% |
| • AC-T regimen | 95% |
| • FAC regimen | 5% |
| **Hormone therapy** | 60% |

**Abbreviations:** AC–T, adriamycin, cyclophosphamide and paclitaxel; FAC, fluorouracil, adriamycin and cyclophosphamide.

**Table 2. Mean values of DVH parameters for heart, LAD, lungs and PTV for patients treated between two techniques.**

| Targets | Parameters | ACSB technique | | FB technique | | P value |
|---|---|---|---|---|---|---|
| | | Mean ± SD | 95% CI | Mean ± SD | 95% CI | |
| **PTV50** | Volume (ml) | 817.55 ± 198.25 | 724.77–910.33 | 809.29 ± 202.05 | 714.73–903.85 | 0.2124[†] |
| | D50% (Gy) | 50.00 ± 0.17 | 49.92–50.08 | 50.00 ± 0.13 | 49.93–50.06 | 0.8761[#] |
| | V47.5 (%) | 96.32 ± 1.10 | 95.80–96.83 | 95.98 ± 0.89 | 95.57–96.40 | 0.0929[#] |
| | V57.5 (%) | 0 ± 0.01 | 0–0.01 | 0.01 ± 0.01 | 0–0.01 | 0.5236[#] |
| **Heart** | Volume (ml) | 576.48 ± 101.56 | 528.95–624.01 | 603.58 ± 115.66 | 549.45–657.70 | 0.0072[#] |
| | Mean dose (Gy) | 9.17 ± 1.25 | 8.59–9.76 | 9.81 ± 1.41 | 9.15–10.47 | <0.0001[†] |
| | Maximum dose (Gy) | 43.79 ± 5.71 | 41.12–46.46 | 45.45 ± 5.15 | 43.04–47.86 | 0.0144[†] |
| | V30 (%) | 1.82 ± 1.85 | 0.95–2.69 | 2.65 ± 2.33 | 1.56–3.74 | 0.0009[†] |
| | V25 (%) | 3.66 ± 2.73 | 2.38–4.93 | 4.85 ± 3.00 | 3.45–6.25 | 0.0011[†] |
| | V20 (%) | 6.88 ± 3.40 | 5.29–8.47 | 8.65 ± 3.37 | 7.07–10.23 | 0.0010[#] |
| | V15 (%) | 12.87 ± 3.85 | 11.07–14.67 | 15.18 ± 4.09 | 13.27–17.09 | 0.0002[†] |
| | V10 (%) | 27.00 ± 6.42 | 24.00–30.01 | 30.91 ± 9.55 | 26.44–35.38 | 0.0012[#] |
| | V5 (%) | 79.18 ± 10.64 | 74.20–84.16 | 81.21 ± 11.06 | 76.04–86.39 | 0.1206[†] |
| **LAD** | Volume (ml) | 2.74 ± 0.88 | 2.32–3.15 | 2.74 ± 0.89 | 2.32–3.15 | 0.9904[†] |
| | Mean dose (Gy) | 19.24 ± 4.79 | 17.00–21.48 | 21.85 ± 5.32 | 19.36–24.34 | 0.0036[†] |
| | Maximum dose (Gy) | 37.87 ± 7.43 | 34.39–41.35 | 39.92 ± 7.78 | 36.28–43.56 | 0.1049[†] |
| | D2% (Gy) | 34.46 ± 7.58 | 30.91–38.01 | 37.33 ± 8.00 | 33.58–41.07 | 0.0174[†] |
| **Ipsilateral lung** | Volume (ml) | 1,056.68 ± 234.37 | 946.99–1,166.37 | 859.94 ± 109.84 | 808.53–911.35 | 0.0007[#] |
| | Mean dose (Gy) | 16.94 ± 2.58 | 15.73–18.14 | 16.70 ± 1.35 | 16.06–17.33 | 0.0793[#] |
| | V20 (%) | 29.84 ± 7.31 | 26.42–33.26 | 28.29 ± 3.05 | 26.86–29.71 | 0.4781[#] |
| | V15 (%) | 39.91 ± 7.68 | 36.31–43.50 | 39.08 ± 3.67 | 37.36–40.80 | 0.1005[#] |
| | V10 (%) | 53.88 ± 7.89 | 50.19–57.57 | 53.19 ± 3.50 | 51.55–54.82 | 0.1454[#] |
| | V5 (%) | 91.15 ± 7.71 | 87.54–94.76 | 93.75 ± 4.99 | 91.41–96.09 | 0.0163[†] |
| **Contralateral lung** | Volume (ml) | 1,372.70 ± 260.37 | 1,250.84–1,494.55 | 1,147.05 ± 167.85 | 1,068.50–1,225.61] | <0.0001[†] |
| | V5 (%) | 15.74 ± 5.76 | 13.04–18.43 | 15.97 ± 5.84 | 13.24–18.71 | 0.2110[#] |
| **Whole lung** | Volume (ml) | 2,426.52 ± 480.70 | 2,201.54–2,651.49 | 2,003.96 ± 259.61 | 1,882.47–2,125.47 | <0.0001[†] |

† Paired–samples t–test was used for normal distribution data.

# Wilcoxon signed ranks test was used for non–normal distribution data.

**Abbreviations:** DVH, dose–volume histogram; PTV, planning target volume.

## Lung volumes and doses

All evaluated lung volumes were statistically significant increased for ACSB compared to FB technique with mean whole lung volumes (2,426.52 vs 2,003.96 ml, p < 0.0001), mean ipsilateral lung volumes (1,056.68 vs 859.94 ml, p = 0.0007) and mean contralateral lung volumes (1,372.70 vs 1,147.05 ml, p < 0.0001). However, other evaluated dose metrics did not show any statistical difference except the mean V5 of the ipsilateral lung significantly lower with ACSB technique (91.15 vs 93.75%, p = 0.002) as seen in Table 2.

## Cardiopulmonary organ volumes and doses correlations

As we found significant differences in cardiopulmonary organ volumes, heart, and LAD dose parameters, we arranged the Pearson correlation coefficient to find the correlation between volume and dose. The result revealed that there is a significant association between increasing of whole lung volume and decreasing of mean heart dose (19.6% by R-squared, p = 0.05) as well as decreasing of mean LAD dose (21.1% by R-squared, p = 0.041) as displayed in Fig 2.

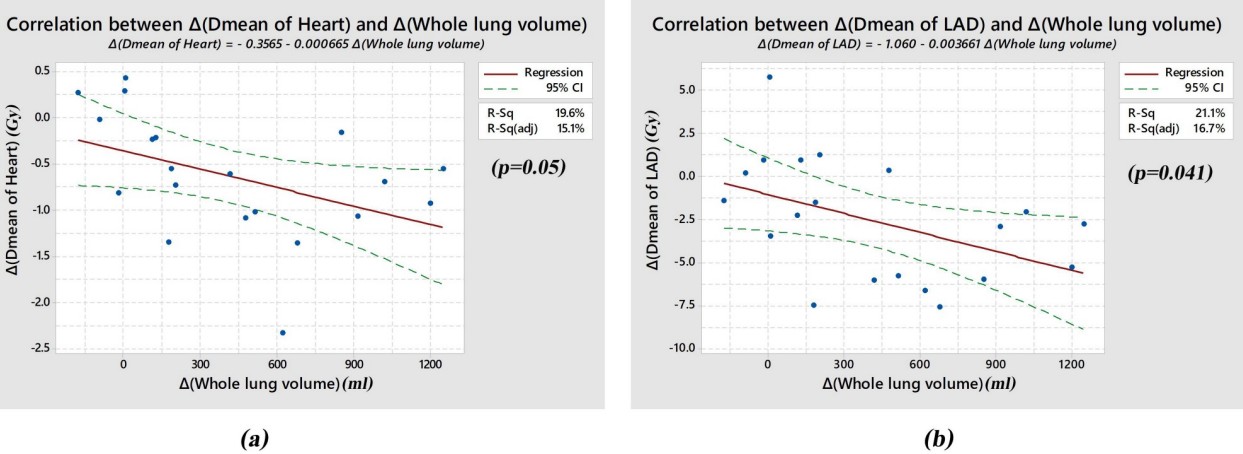

**Fig 2.** Correlation Between the Differences in Mean Heart Dose and Whole Lung Volume (a) and the Differences in Mean LAD Dose and Whole Lung Volume (b).

## Discussion

In an attempt to reduce cardiac dose in LtBC radiotherapy, the consensus guidelines recommend providing radiation fields with the exclusion of as much of the heart as possible to minimize the risk of heart disease without compromising target coverage [7]. From the aforementioned principle of using DIBH to reduce cardiopulmonary dose in LtBC treatment, this present study demonstrated the new technique called ACSB that applied abdominal compressor during a patient being in deep inspiration phase and assisted the patient to carry on the shallow breathing during simulation and treatment to create negative intrathoracic pressure as much as possible. Under physiological conditions, the negative intrathoracic pressure not only briefly facilitates the venous return to the right side of the heart but also distends extra-alveolar vascular structures. Therefore, the blood tends to be retained in the right ventricle or the lung and decreased the venous return to the left side of the heart. The hemodynamic consequence is decreasing of the left ventricular end-diastolic volume, end-systolic volume, and stroke volume while increasing of lung volume [27]. Accordingly, this will be caused of decreasing the heart volume and increasing the distance between heart and chest wall as seen from using our ACSB technique (Fig 3). Additionally, the ACSB technique is suitable for our center that do not have beam gating system to carry on the DIBH procedure.

When compared to the FB technique, the ACSB offering statistically significant decreased in the volume of the heart treated and most of dosimetric parameters of the heart. Our current results were supported by a significant relationship between mean heart dose and mean LAD dose reduction and expansion of whole lung volume as seen in Fig 2. This significant correlation was 19.6% and 21.1% by R-squared with $p \leq 0.05$, showing the more whole lung volume was expanded, the more reduction of mean heart dose and mean LAD dose were created.

One systematic review [19] reporting heart doses for different radiotherapy techniques in LtBC radiotherapy revealed a MHD of 8.6 Gy from the IMRT technique. Further, other LtBC treatment studies using Helical TomoTherapy [28, 29] reported the MHD in the range of 8.6–12.2 Gy without using any respiratory management. These results are consistent with MHD from both ACSB and FB techniques in our research. In summary, a 7% reduction of the mean of MHD and 11.95% reduction of the mean of mean LAD dose were found using our ACSB technique (Fig 4). Notably, this benefit is quite lower than other retrospective studies of modified DIBH in which the value ranged from 18–62% [10–18] because our technique performs

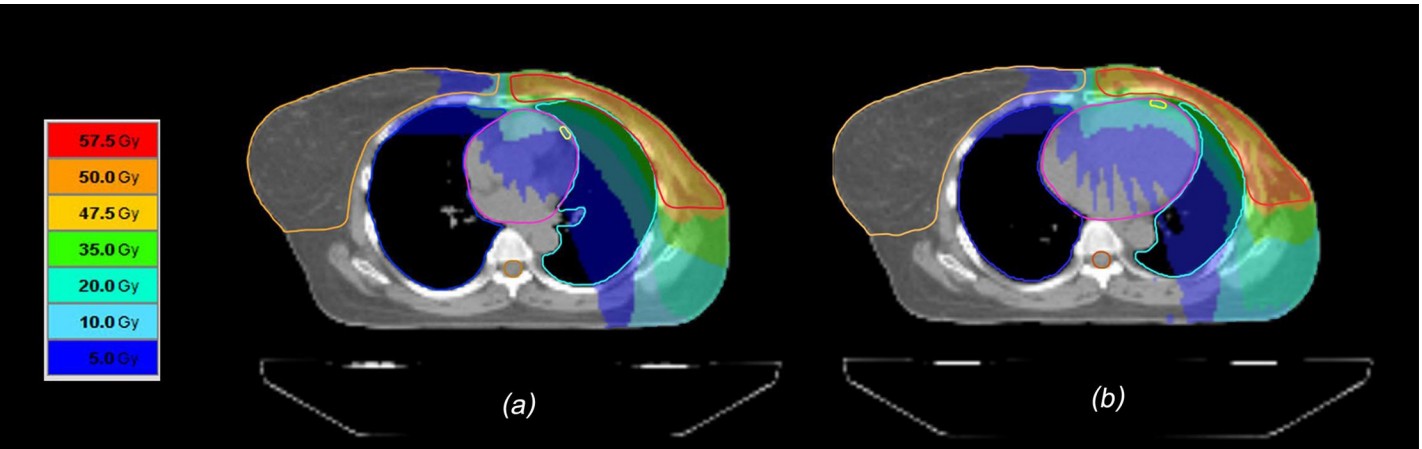

**Fig 3.** Comparison of Dose Distributions with ACSB (a) and FB Technique (b).

only shallow breathing, not a full deep inspiration. Besides, this variable could have resulted from the differences in treatment volumes, radiation techniques, or the technique of planning as well. However, according to a report by Darby et al. [5] who did a population-based case–control study in a cohort of 2158 women underwent breast cancer radiotherapy to look at major coronary events and ischemic cardiac deaths. They revealed that major coronary events increased linearly with the mean heart dose delivered by 7.4% per Gy. This increase began within 5 years of treatment regardless of current cardiac risk factors at the time of radiotherapy. Therefore, with only a small dose reduction (0.64 Gy of the mean of the MHD reduction) from our ACSB technique, it could have at least clinical benefit to reduce the risk of major coronary events when compare with FB technique.

Despite the significant increase of lung volume with the ACSB technique, the evaluated dose parameters of the lung were comparable except for the V5 of the ipsilateral lung. Walston et al. [18] proposed that mean dose and V20 of the ipsilateral and total lung volume were not significantly reduced by the DIBH technique from their large clinical series which is compatible with our lung dose results. We supposed that most of the lung volume treated was not changed from the expansion of the whole lung due to a similar relative volume being included

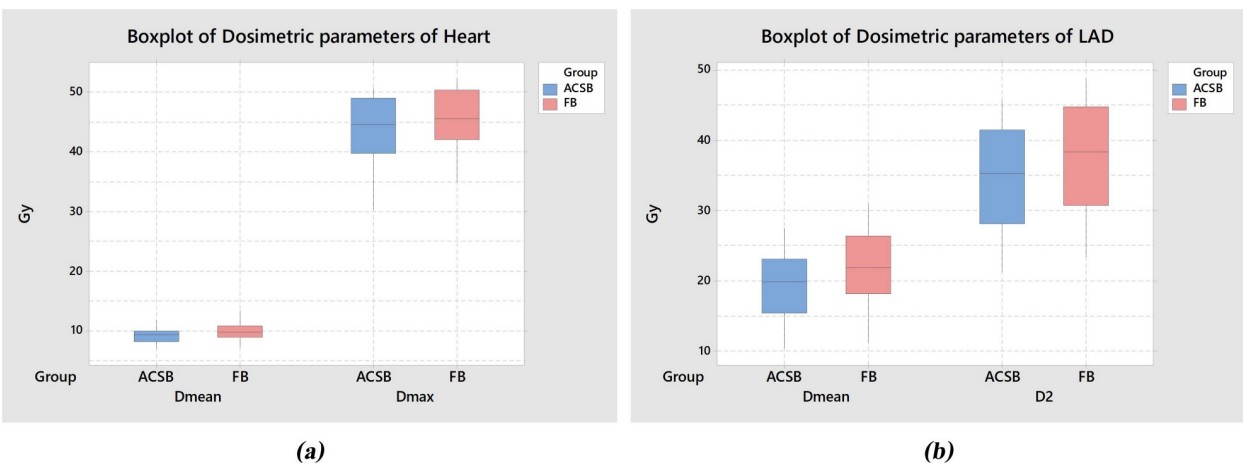

**Fig 4.** Boxplots of Dosimetric Parameters of Heart (a) and LAD (b) of 20 Patients.

in the radiated area consequently most of our lung dose parameters did not show any statistically significant differences. Nonetheless, the other advantages of using the ACSB technique were stabilization and prolongation of shallow breathing. Also, it could reduce the target motion that is superior in target dose coverage during treatment to overcome the limitation of flash function when treating by HT as our center.

Our ACSB technique exhibited its effectiveness in cardiac sparing for PMRT in LtBC. Although this procedure applied abdominal compression to help patients to keep shallow respiration, good compliance is still required from the patient. Other concerned points for our technique are about a time and consistency of doing shallow breathing during treatment delivery. Due to lacking gating and tracking system in our center, we have planned to measure the duration that each patient can perform a consistent shallow breathing during CT simulation by monitoring from respiratory graph. We will use this information to select the patient who is able to tolerate for ACSB technique. We think this process is simple and can be reproduced in all centers with limited resource. Nevertheless, even with using image-guided radiotherapy (IGRT) for the treatment delivery, the discordant between calculated dose and delivered dose still can happen. We will improve our ACSB technique by adding the 4D-CT simulation in our process and creating the internal target volume (ITV) to account for respiratory organ movement before planning. Therefore, further research would have more intensive patient coaching about deep and shallow breathing, and we will also report the success and issue of using the ACSB technique in real life practice.

## Conclusion

A new procedure for cardiac sparing radiotherapy of PMRT in left breast cancer was observed by using an abdominal compression with shallow breathing from our single-institution prospective study. Significant reduction in heart and LAD dose can be achieved compared to the free breathing technique. The ACSB technique is simple, well-tolerated, and can be applied in the limited resource center.

## Supporting information

**S1 Table. Dose constraints to the PTV and OARs.**
(DOCX)

## Author Contributions

**Conceptualization:** Pitchayaponne Klunklin, Somsak Wanwilairat, Imjai Chitapanarux.

**Data curation:** Chalardchay Pratoomchart, Wannapha Nobnop.

**Formal analysis:** Kittikun Kittidachanan.

**Investigation:** Chalardchay Pratoomchart, Pitchayaponne Klunklin, Wannapha Nobnop.

**Methodology:** Pitchayaponne Klunklin, Somsak Wanwilairat.

**Supervision:** Imjai Chitapanarux.

**Writing – original draft:** Chalardchay Pratoomchart, Pitchayaponne Klunklin.

**Writing – review & editing:** Pitchayaponne Klunklin, Wannapha Nobnop, Imjai Chitapanarux.

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
