## [Decision Letter · Decision Letter 0]

7 Apr 2021

PONE-D-21-05183

The advantages of abdominal compression with shallow breathing during left-sided postmastectomy radiotherapy by Helical TomoTherapy

PLOS ONE

Dear Dr. Klunklin,

Thank you for submitting your manuscript to PLOS ONE. After careful consideration, we feel that it has merit but does not fully meet PLOS ONE’s publication criteria as it currently stands. Therefore, we invite you to submit a revised version of the manuscript that addresses the points raised during the review process.

We look forward to receiving your revised manuscript.

Kind regards,

Jennifer Wei Zou, Ph.D.

Academic Editor

PLOS ONE

Journal Requirements:

2) In your Methods section, please provide additional information about the participant recruitment method and the demographic details of your participants. Please ensure you have provided sufficient details to replicate the analyses such as:

a) the recruitment date range (month and year),

b) a description of any inclusion/exclusion criteria that were applied to participant recruitment,

c) a statement as to whether your sample can be considered representative of a larger population, and

d) a description of how participants were recruited."

3) Please provide additional details regarding participant consent. In the ethics statement in the Methods and online submission information, please ensure that you have specified whether consent was informed.

Reviewers' comments:

Reviewer's Responses to Questions

**Comments to the Author**

1. Is the manuscript technically sound, and do the data support the conclusions?

Reviewer #1: Yes

Reviewer #2: Partly

2. Has the statistical analysis been performed appropriately and rigorously? 

Reviewer #1: Yes

Reviewer #2: No

3. Have the authors made all data underlying the findings in their manuscript fully available?

Reviewer #1: Yes

Reviewer #2: Yes

4. Is the manuscript presented in an intelligible fashion and written in standard English?

Reviewer #1: Yes

Reviewer #2: Yes

5. Review Comments to the Author

Reviewer #1: The manuscript described a compromising method to spare cardiopulmonary organs during left-sides post-mastectomy radiotherapy (PMRT). That is, at the situation of lacking a beam gating system, through relaxing the deep-inspiration breath-hold (DIBH) technique into abdominal compression with shallow breathing (ACSB), the cardiopulmonary dose could still be possibly reduced, when comparing to the free breathing mode. The paper is relatively clearly written, yet, I suggest further clarifications on several issues of the manuscript.

Main comments:

A major concern of the proposed ACSB method is its dose calculation. In the DIBH or irradiating at a specific respiratory phase through gating, the dose calculation will not be affected by the lung motion. However, in the proposed ACSB method, although shallow breathing, the calculated dose based on the simulation CT may differ from the delivered dose to the patients. Can the author discuss this or if a 4DCT based dose calculation can be performed?

In the discussion section, the authors compared the heart dose sparing via ACSB to other radiotherapy techniques. Obviously, the dose reduction from ACSB is much smaller than those previously reported. This weakens the potential impact of the research work. Considering this, I would suggest the authors discuss quantitatively the correlation between heart dose degradation and the reduction of cardiopulmonary side effect probabilities, such that it can be clearer whether or not the proposed ACSB method has practical clinical values.

Minor comments:

1. First sentence of the last paragraph in the “introduction” section, “According to the principle…of the thorax”. The sentence is hard to follow. Please revise.

2. In the section of “Statistical Analysis”, the definitions of δ and ρ are not stated before they are used.

3. Figures 2 and 4 are of low resolution. Please consider a replacement.

Reviewer #2: The reviewer thinks the research topic of using abdominal compression to facilitate shallow breathing is interesting and clinically useful for facilities not having gating capability during radiation treatment delivery. Please see comments below.

1. It is mentioned in the article “When the respiratory cycle graph showed a normal respiratory curve (Fig 1) which indicated that the patient was breathing regularly, the CT scan was undertaken.” However the authors did not describe the treatment delivery condition. Has any patient fail to be compliant during treatment delivery and how was it managed? The reviewer expect the author to include more details, for example, the average breathing motion range and other practical aspects such as time required for scan versus treatment delivery and if patient can maintain consistent shallow breathing through treatment. Readers would be interested in knowing if this technique was able to be applied to daily treatment and if any concerns when being used.

2. Please provide more details and formula used to estimate the sample size of 21 mentioned in the first paragraph of statistical analysis.

3. “A physiologic increase in heart volume was noted with the ACSB technique.” This disagrees with table. Please clarify.

4. The first paragraph in section “Heart and LAD Volumes and Doses” summarizes the comparison between ACSB and FB. The correlation between lung volume and heart/LAD doses were presented. The following paragraph “Under physiological conditions, this negative intrathoracic pressure not…… keep the benefit of deep inspiration.” tried to qualitatively explain why ACSB results in some lower metrics for the heart and LAD. However, it is very hard to read. Please re-write and elaborate.

5. “Under physiological conditions, this negative intrathoracic pressure not……and stroke volume are decrease.” Please provide reference for the paragraph.

6. Please clarify in Table 2 why some statistical tests were t-test and the others were Wilcoxon.

7. The language can be improved. For example, and not limited to, the underlined sentences:

“For the OARs, at least one of the mandatory constraints needed to meet the criteria.” It sounds misleading as if only one constraint out of all 6 OARs needs to meet, which is unlikely. Maybe change it to “For each OAR,…”

“For two process, DIBH….”

“According to the principle of …that will ….escalate….”

8. The quality of figures are extremely poor. Please improve on the resolution.

6. PLOS authors have the option to publish the peer review history of their article (what does this mean?). If published, this will include your full peer review and any attached files.

Reviewer #1: No

Reviewer #2: No

---

## [Author Response · Author response to Decision Letter 0]

21 Apr 2021

April 19, 2021

Dear Dr. Jennifer Wei Zou and reviewers,

We are very excited to have been given the opportunity to submit a revised draft of my manuscript titled “The advantages of abdominal compression with shallow breathing during left-sided postmastectomy radiotherapy by Helical TomoTherapy” to PLOS ONE. I am grateful to the editor and reviewers for taking the time and effort necessary to provide such insightful guidance on my manuscript. We carefully considered all the comments offered by your team. I have highlighted the changes within the manuscript based on those comments. Here is a point-by-point response to the reviewers’ comments and recommendations.

Editor’s comment

Response: We already reviewed and corrected our manuscript style followed The PLOS ONE style templates.

2) In your Methods section, please provide additional information about the participant recruitment method and the demographic details of your participants. Please ensure you have provided sufficient details to replicate the analyses such as:

a) the recruitment date range (month and year),

b) a description of any inclusion/exclusion criteria that were applied to participant recruitment,

c) a statement as to whether your sample can be considered representative of a larger population, and

d) a description of how participants were recruited."

Response: We appreciate the Editor’s comments and added following statements in the methods section. 

- edit to the manuscript can be found in the “Patients” sub-section on page 4-5 with green highlight.

3) Please provide additional details regarding participant consent. In the ethics statement in the Methods and online submission information, please ensure that you have specified whether consent was informed.

Response: We thank the Editor for the feedback. We asserted information about consent in Methods section. 

- The information can be found in the “Patients” sub-section, page 4, line 91-93 with green highlight.

Reviewer 1: 

The manuscript described a compromising method to spare cardiopulmonary organs during left-sides post-mastectomy radiotherapy (PMRT). That is, at the situation of lacking a beam gating system, through relaxing the deep-inspiration breath-hold (DIBH) technique into abdominal compression with shallow breathing (ACSB), the cardiopulmonary dose could still be possibly reduced, when comparing to the free breathing mode. The paper is relatively clearly written, yet, I suggest further clarifications on several issues of the manuscript.

Main comments:

1) A major concern of the proposed ACSB method is its dose calculation. In the DIBH or irradiating at a specific respiratory phase through gating, the dose calculation will not be affected by the lung motion. However, in the proposed ACSB method, although shallow breathing, the calculated dose based on the simulation CT may differ from the delivered dose to the patients. Can the author discuss this or if a 4DCT based dose calculation can be performed?

Response: We agree with your suggestion. Firstly, we have to clarify that in our research, we only did the ACSB technique in CT simulation process and did not perform it in real treatment. Recently, we are planning to use ACSB in real treatment with image-guided radiotherapy (IGRT) by CBCT. Even with IGRT, the discordant between calculated dose and delivered dose still can happen as you concerned. According to your advice, we will add the 4D-CT simulation in ACSB process and create the ITV to account for respiratory organ movement before planning. 

- We also added this comment in the discussion part. (page 13, line 274-278 with yellow highlight)

2) In the discussion section, the authors compared the heart dose sparing via ACSB to other radiotherapy techniques. Obviously, the dose reduction from ACSB is much smaller than those previously reported. This weakens the potential impact of the research work. Considering this, I would suggest the authors discuss quantitatively the correlation between heart dose degradation and the reduction of cardiopulmonary side effect probabilities, such that it can be clearer whether or not the proposed ACSB method has practical clinical values.

Response: Thank you and we agree with your suggestion.

 Regarding to the previous study reported in 2013 (Darby SC, Ewertz M, McGale P, et al. Risk of ischemic heart disease in women after radiotherapy for breast cancer. N Engl J Med. 2013;368(11):987-998.), they proposed that the rates of major coronary events increased linearly with the mean dose to the heart by 7.4% per gray. Though with a small dose reduction (0.64 Gy of MHD reduction) from our ACSB technique, it could have at least clinical benefit to reduce the risk of major coronary event when compare with FB technique. 

- We inserted this discussion with reference (no.30) into the discussion part. (page 12, line 247-254 with yellow highlight)

Minor comments:

1) First sentence of the last paragraph in the “introduction” section, “According to the principle…of the thorax”. The sentence is hard to follow. Please revise.

Response: Thank you very much for your advice. After carefully read, we had rewritten this complicated sentence as your suggestion.

- Introduction section, page 4, line 80-82 with yellow highlight

2) In the section of “Statistical Analysis”, the definitions of δ and ρ are not stated before they are used.

Response: The definitions of δ and ρ are now stated in the statistical analysis part as suggested.

- “Statistical analysis” sub-section, page 6, line 145-151 with yellow highlight

3) Figures 2 and 4 are of low resolution. Please consider a replacement.

Response: Thank you for highlighting this issue. We have formatted all the figure files in line with journal requirements and have used the PACE platform to now generate a .tif version of the file. We have included the .tif file alongside the revised manuscript.

Reviewer 2: 

The reviewer thinks the research topic of using abdominal compression to facilitate shallow breathing is interesting and clinically useful for facilities not having gating capability during radiation treatment delivery. Please see comments below.

1) It is mentioned in the article “When the respiratory cycle graph showed a normal respiratory curve (Fig 1) which indicated that the patient was breathing regularly, the CT scan was undertaken.” However the authors did not describe the treatment delivery condition. Has any patient fail to be compliant during treatment delivery and how was it managed? The reviewer expect the author to include more details, for example, the average breathing motion range and other practical aspects such as time required for scan versus treatment delivery and if patient can maintain consistent shallow breathing through treatment. Readers would be interested in knowing if this technique was able to be applied to daily treatment and if any concerns when being used.

Response: Thank you for highlighting this issue. Firstly, we have to clarify that in our research, we only did the ACSB technique in CT simulation process and did not perform it in actual treatment yet. As you have the concerned points about time and consistency of doing shallow breathing, due to lacking gating and tracking system in our center, we planned to measure the duration which each patient can do a consistent shallow breathing during CT simulation by monitoring from respiratory graph. We will use this information to select the patient who is able to tolerate for ACSB technique. We think this process is simple and can be reproduced in all centers with limited resource.

- We added this clarification in the discussion part, page 13, line 269-274 with blue highlight

2. Please provide more details and formula used to estimate the sample size of 21 mentioned in the first paragraph of statistical analysis.

Response: Thank you for your advice. We have added more detail about sample size calculation along with the reference as suggested. 

- “Statistical analysis” sub-section, page 6-7, line 143-155 with blue highlight with refence no.26

3. “A physiologic increase in heart volume was noted with the ACSB technique.” This disagrees with table. Please clarify.

Response: We are grateful to the reviewer for noticing this error and already corrected the sentence.

- Result section, page 9, line 182 with blue highlight

4. The first paragraph in section “Heart and LAD Volumes and Doses” summarizes the comparison between ACSB and FB. The correlation between lung volume and heart/LAD doses were presented. The following paragraph “Under physiological conditions, this negative intrathoracic pressure not…… keep the benefit of deep inspiration.” tried to qualitatively explain why ACSB results in some lower metrics for the heart and LAD. However, it is very hard to read. Please re-write and elaborate.

Response: Thank you for your comment. We already revised this paragraph.

- Discussion part, page 11, line 221-227 with blue highlight

5. “Under physiological conditions, this negative intrathoracic pressure not……and stroke volume are decrease.” Please provide reference for the paragraph.

Response: We added the requested reference for this paragraph (ref no. 27) 

6. Please clarify in Table 2 why some statistical tests were t-test and the others were Wilcoxon.

Response: We chose the statistical method depended on type of data. After data gathering, we did the normality test after that we used the sample t-test for normal distribution data while used the Wilcoxon signed ranks test for non-normal distribution data. We also added this description in the table legend of Table 2.

- Edit can be found at the table legend below Table 2.

7. The language can be improved. For example, and not limited to, the underlined sentences:

“For the OARs, at least one of the mandatory constraints needed to meet the criteria.” It sounds misleading as if only one constraint out of all 6 OARs needs to meet, which is unlikely. Maybe change it to “For each OAR,…”

“For two process, DIBH….”

“According to the principle of …that will ….escalate….”

Response: We attempted to improve our language in some sentences. Please find the edit sentence in revise version.

- Introduction section, page 3, line 61-62 

- Introduction section, page 4 line 80-82

- Method section, page 6, line 138

8. The quality of figures are extremely poor. Please improve on the resolution.

Response: Thank you for highlighting this issue. We have formatted all of the figure files in line with journal requirements and have used the PACE platform to now generate a .tif version of the file. We have included the .tif file alongside the revised manuscript.

Sincerely,

Pitchayaponne Klunklin, MD

---

## [Decision Letter · Decision Letter 1]

16 Jun 2021

PONE-D-21-05183R1

The advantages of abdominal compression with shallow breathing during left-sided postmastectomy radiotherapy by Helical TomoTherapy

PLOS ONE

Dear Dr. Klunklin,

Thank you for submitting your manuscript to PLOS ONE. After careful consideration, we feel that it has merit but does not fully meet PLOS ONE’s publication criteria as it currently stands. Therefore, we invite you to submit a revised version of the manuscript that addresses the points raised during the review process.

We look forward to receiving your revised manuscript.

Kind regards,

Jennifer Wei Zou, Ph.D.

Academic Editor

PLOS ONE

Journal Requirements:

Additional Editor Comments (if provided):

From Reviewer 1:

I am happy to see the authors have addressed my questions. Some minor problems:

1. Line 153-155, it is not clear to me how the sample size of 21 is determined with using various \\rho values. Please specify.

2. Line 183-184, ' Moreover, using ACSB resulted in a significant decrease in mean heart dose (MHD) with 9.17 vs 1839.81 Gy (p<0.0001) and maximum heart dose with 43.79 vs 45.45 Gy (p=0.0144).' I think here more accurately, the 'mean heart dose' should be the mean of the mean heart dose and the 'maximum heart dose' is 'the mean of the maximum heart dose'.

3. Line 187, 'no difference in the LAD volume' should be 'no statistical difference in the mean LAD volume'. There are some other places with similar problems that I would suggest the authors consider for a more accurate expression.

4. Figure 2, please add units to x and y axis labels.

Reviewers' comments:

Reviewer's Responses to Questions

**Comments to the Author**

1. If the authors have adequately addressed your comments raised in a previous round of review and you feel that this manuscript is now acceptable for publication, you may indicate that here to bypass the “Comments to the Author” section, enter your conflict of interest statement in the “Confidential to Editor” section, and submit your "Accept" recommendation.

Reviewer #1: All comments have been addressed

Reviewer #2: All comments have been addressed

2. Is the manuscript technically sound, and do the data support the conclusions?

Reviewer #1: Yes

Reviewer #2: Yes

3. Has the statistical analysis been performed appropriately and rigorously? 

Reviewer #1: N/A

Reviewer #2: Yes

4. Have the authors made all data underlying the findings in their manuscript fully available?

Reviewer #1: Yes

Reviewer #2: Yes

5. Is the manuscript presented in an intelligible fashion and written in standard English?

Reviewer #1: Yes

Reviewer #2: Yes

6. Review Comments to the Author

Reviewer #1: (No Response)

Reviewer #2: Author has addressed all points accordingly in their revision. The reviewer has no further comments.

7. PLOS authors have the option to publish the peer review history of their article (what does this mean?). If published, this will include your full peer review and any attached files.

Reviewer #1: No

Reviewer #2: No

---

## [Author Response · Author response to Decision Letter 1]

21 Jun 2021

June 21, 2021

Dear Dr. Jennifer Wei Zou and reviewers,

Thank you very much for you kindly suggestion. We carefully considered all the comments. I have highlighted the changes within the manuscript based on those comments by using “Track Changes” option in Microsoft Word. Here is a point-by-point response to the reviewer’s comments and recommendations.

Journal Requirements:

Response: We are thankful for your suggestion. 

- We found that the reference no. 30 is a duplicated of no.5, therefore; we deleted the reference no. 30 and corrected the reference no. within the main text. (Line 249)

- The reference no. 20 is only an abstract. We changed the reference style followed the Vancouver referencing style for abstract. (Line 343-347)

- Few corrections of format of the reference no. 10 and 14.

From Reviewer 1:

1. Line 153-155, it is not clear to me how the sample size of 21 is determined with using various \\rho values. Please specify.

Response: As we clarified that we do not have an information about but we believed that the correlation should be more than 0.5 because we planned to measure the radiation dose in the same individual. We calculated the sample size with from 0.5-0.8 (showed in the table) and selected to use the highest sample size number from lung using 0.5. 

We also added more information about sample size calculation into statistical analysis subsection (Line 153-156)

Heart

SD=1.4 

N

 0.5 11

 0.6 9

 0.7 6

 0.8 4

Lung

SD=2.02 

N

 0.5 21

 0.6 16

 0.7 12

 0.8 8

2. Line 183-184, ' Moreover, using ACSB resulted in a significant decrease in mean heart dose (MHD) with 9.17 vs 1839.81 Gy (p<0.0001) and maximum heart dose with 43.79 vs 45.45 Gy (p=0.0144).' I think here more accurately, the 'mean heart dose' should be the mean of the mean heart dose and the 'maximum heart dose' is 'the mean of the maximum heart dose'.

Response: Thank you for your comment. We modified the manuscript followed your recommendation.

- Line 184-185 

- Line 244-245

- Line 254

3. Line 187, 'no difference in the LAD volume' should be 'no statistical difference in the mean LAD volume'. There are some other places with similar problems that I would suggest the authors consider for a more accurate expression.

Response: Thank you and we agree with your suggestion. We already made a change of our manuscript as your suggestion. 

- Line 189

- Line 199

- Line 234

4. Figure 2, please add units to x and y axis labels.

Response: We are grateful to the reviewer for noticing this error and already add units of the x and y axis in figure 2.

Sincerely,

Pitchayaponne Klunklin, MD

---

## [Editor Report · Decision Letter 2]

7 Jul 2021

The advantages of abdominal compression with shallow breathing during left-sided postmastectomy radiotherapy by Helical TomoTherapy

PONE-D-21-05183R2

Dear Dr. Klunklin,

We’re pleased to inform you that your manuscript has been judged scientifically suitable for publication and will be formally accepted for publication once it meets all outstanding technical requirements.

Kind regards,

Jennifer Wei Zou, Ph.D.

Academic Editor

PLOS ONE

Additional Editor Comments (optional):

The authors have addressed the reviewers' comments. The manuscript is accepted.
---

## [Editor Report · Acceptance letter]

9 Jul 2021

PONE-D-21-05183R2 

The advantages of abdominal compression with shallow breathing during left-sided postmastectomy radiotherapy by Helical TomoTherapy 

Dear Dr. Klunklin:

I'm pleased to inform you that your manuscript has been deemed suitable for publication in PLOS ONE. Congratulations! Your manuscript is now with our production department. 

Kind regards, 

on behalf of

Dr. Jennifer Wei Zou 

Academic Editor

PLOS ONE